# Ångström exponent impact on the aerosol optical properties obtained from vibrational-rotational Raman lidar observations

Gladiola Malollari[1], Albert Ansmann[2], Holger Baars[2], Cristofer Jimenez[2], Julian Hofer[2], Ronny Engelmann[2], Nathan Skupin[2], and Seit Shallari[1]

[1]Agricultural University of Tirana, Tirana, Albania
[2]Leibniz Institute for Tropospheric Research, Leipzig, Germany

**Correspondence:** Gladiola Malollari (gmalollari@ubt.edu.al)

**Abstract.** Vertical profiles of aerosol properties are essential for assessing the impact of aerosols on cloud formation and the Earth's radiation budget. Lidars can provide profiles of the particle backscatter and extinction coefficients and the extinction-to-backscatter ratio (lidar ratio). An Ångström exponent has to be assumed when computing these profiles from nitrogen vibrational-rotational Raman signals. This assumption introduces uncertainties. An alternative approach is the rotational Raman
lidar method, which does not need an Ångström exponent as input. This study presents a quantitative comparison between the pure rotational and vibrational-rotational Raman lidar approaches to assess the impact of the Ångström exponent assumption on the vibrational-rotational Raman lidar solutions. In this short article, we present four contrasting case studies based on observations of wildfire smoke, Saharan dust, residential wood combustion smoke, and a cirrus layer. The optical properties are derived at a wavelength of 532 nm, with the rotational Raman signals measured at 530 nm and the vibrational-rotational
Raman signals measured at 607 nm. It was found that the use of an Ångström exponent, deviating by 1 from the true value, introduces relative uncertainties of 5% and less (backscatter coefficient), 5-10% (extinction coefficient), and around 10% (lidar ratio) in the vibrational-rotational Raman lidar solutions.

## 1 Introduction

Aerosols can influence the radiation balance of the Earth–atmosphere system directly by scattering and absorbing longwave
and shortwave radiation and indirectly by influencing cloud formation and precipitation. Lidar (LIght Detection And Ranging) instruments enable vertical profiling of the aerosol optical properties. Height-resolved independent measurements of the aerosol backscatter coefficient, extinction coefficient, and thus the lidar ratio are possible with an HSRL (high-spectral-resolution lidar) (Eloranta, 2005) and Raman lidars (Ansmann et al., 1990, 1992). Only HSRL and Raman lidars allow the direct determination of particle extinction height profiles (Ansmann and Müller, 2005).

HSRL and rotational Raman lidars do not need any assumption on the wavelength dependence of aerosol extinction in the determination of the particle extinction profile at the laser wavelength, e.g., at 532 nm. In contrast, the widely used nitrogen rotational-vibrational Raman lidar technique needs to assume an Ångström exponent (A), which describes the wavelength dependence of aerosol extinction. In the case of a 532 nm nitrogen vibrational-rotational Raman lidar, the laser light is attenuated

on the way to the backscatter region at 532 nm and, after the Raman scattering process, on the way back to the lidar at 607 nm.

Overestimation or underestimation of the Ångström exponent leads to an error in the derived profile of the 532 nm particle extinction coefficient. In the case of a rotational Raman lidar, the laser light is attenuated on the way from the backscatter region back to the lidar at 530 nm (close to the transmitted wavelength) so that an Ångström exponent value is not needed as input in the extinction retrieval (Veselovskii et al., 2015; Zenteno-Hernández et al., 2021). Veselovskii et al. (2015) and Zenteno-Hernández et al. (2021) showed comparisons of extinction profiles obtained with the rotational Raman and vibrational-rotational Raman

lidar method for a typical Ångström exponent of A=1 in the extinction retrieval from the vibrational-rotational Raman signals and found good agreement. However, an extended investigation of the full impact of an erroneous A input on the accuracy of the set of backscatter, extinction, and lidar ratio solutions obtained from vibrational-rotational Raman lidar observations has not been presented yet.

In this work, we will show the results of such an extended uncertainty study. Our study is based on observations with a

Raman lidar instrument that was deployed in the aerosol-rich environment of the city of Tirana, Albania. The single-wavelength Raman lidar instrument was equipped with a rotational Raman channel at 530 nm and a vibrational-rotational Raman channel at 607 nm. The data analysis is applied to three contrasting aerosol events with fresh and aged fire smoke and Saharan dust.

This paper is organized as follows: in Section 2 a description of the lidar instrument setup is given. The calculation of the particle backscatter and extinction coefficients from rotational and vibrational-rotational Raman signals is presented as well.

The three distinct aerosol scenarios and a cirrus measurement are presented in Section 3. Concluding remarks are given in Section 4.

## 2 Data and Methods

### 2.1 Polly Raman Lidar

Continuous Raman lidar observations were performed at Tirana, Albania, from November 2022 to October 2023 (Polly, 2025).

A sketch of the lidar system is shown in Fig. 1. The lidar, called POLLY_1v2 (**PO**rtab**L**e Raman **L**idar s**Y**stem), emits linearly polarized light pulses at 532 nm wavelength and records the backscattered radiation from aerosols and molecules in five different channels: the total elastic-backscatter channel at 532 nm, the rotational Raman channel at 530 nm, the vibrational-rotational Raman channel at 607 nm, the cross-polarized elastic-backscatter channel at 532 nm (indicated by 532c in Fig. 1), and the co- or parallel-polarized elastic-backscatter channel at 532 nm (indicated by 532p in Fig. 1) (Althausen et al., 2004, 2009).

The laser head generates pulses with an energy of 120 mJ at 532 nm wavelength. The pulse's repetition rate is 15 Hz (Baars et al., 2016). A 12x beam expander expands the laser beam to a diameter of 60 mm and reduces its divergence to < 0.2 mrad. Before reaching the beam expander, a shutter is mounted to stop the measurements in case of rain or when an aircraft crosses the lidar. The beam is directed into the atmosphere by means of two mirrors. Scattering and absorption of laser light by aerosol particles and air molecules attenuate the laser beam. The scattered light at an angle of 180° is collected by a Newtonian

telescope with a primary mirror of 20 cm in diameter. The backscattered photons go through a pinhole and reach a collimation lens. After passing this lens, the light is reflected by a mirror at 90 degrees. By using dichroic and polarizing beam splitters,

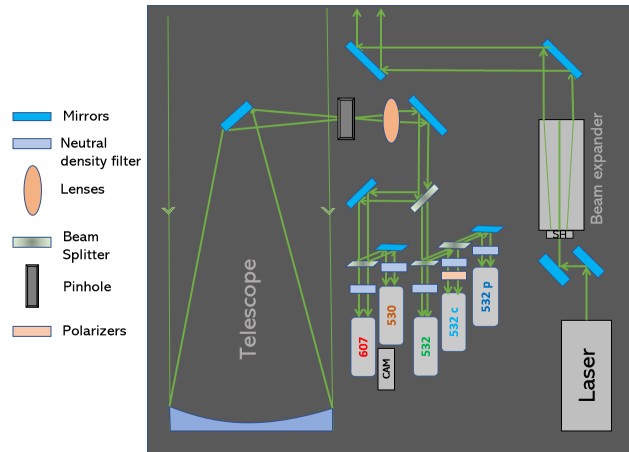

**Figure 1.** The optical setup of the Polly instrument deployed in Tirana, Albania. The numbers indicate the wavelengths of the photon-counting channels. Cross and co-polarized signal channels are indicated by index c and index p, respectively. The camera (CAM) is used to optimize the laser-beam receiver-field-of-view overlap.

the different signal components are separated. A camera is mounted to monitor the overlap between the laser beam and the telescope. According to our analyses, the full overlap between the laser and receiver field of view is achieved at 600 m. Neutral density filters of different attenuation strengths are used to avoid the overloading of the photomultipliers at near-range heights.

Independent measurements of the backscatter coefficient, extinction coefficient, lidar ratio, and depolarization at 532 nm are possible with this instrument by applying the Raman lidar method (Ansmann et al., 1992). Signals measured with rotational and vibrational-rotational Raman channels can be used. Lidar signals are influenced by particle and Rayleigh backscatter and light-extinction processes (Baars et al., 2012). For the removal of Rayleigh scattering effects, molecular backscatter and extinction coefficients are computed using temperature and pressure profiles from GDAS (Global Data and Assimilation System) (Baars

et al., 2012; Bucholtz, 1995). Polly_1v2 is a well-calibrated and adjusted lidar system and permits the retrieval of the particle backscatter profiles at heights above about 150-200 m (Raman lidar method). The computation of the particle backscatter coefficient profile requires a reference value. The calibration of signals is done by choosing a reference value at an altitude where only molecules contribute to the measured signal. The reference height is typically in the upper troposphere or lower stratosphere. Vertical smoothing is applied to the profiles to reduce the noise. Retrieval uncertainties are of the order of 10%

(backscatter coefficient, depolarization ratio), 20% (extinction coefficient), and 25%(lidar ratio) in the aerosol and cirrus studies presented in Section 3.

## 2.2 Particle extinction coefficient

The particle extinction coefficient $\alpha$ can be directly retrieved by using the following equation:

$$\alpha_{\lambda_0}^{\text{par}}(R) = \frac{\frac{\mathrm{d}}{\mathrm{d}R} \ln\left(\frac{N(R)\,O(R)}{P_{\lambda_{\text{Ra}}}(R)R^2}\right) - \alpha_{\lambda_0}^{\text{mol}}(R) - \alpha_{\lambda_{\text{Ra}}}^{\text{mol}}(R)}{1 + \left(\frac{\lambda_0}{\lambda_{\text{Ra}}}\right)^{\text{å}(R)}} \tag{1}$$

where $\lambda_0$ is the laser wavelength (532 nm) and $\lambda_{\text{Ra}}$ is the wavelength (607 nm) of the nitrogen vibrational-rotational Raman channel. The rotational Raman signals are measured at 530 nm. $P_{\lambda_{\text{Ra}}}$ is the received inelastic Raman backscattering signal, and R is the range or height. O(R) is the overlap function, and N(R) is the molecular density profile (Ansmann et al., 1992). The Ångström exponent å(R) must be assumed.

If the extinction coefficient is computed by using the rotational Raman channel $\lambda_{\text{Ra}} = 530$ nm, the ratio $\frac{\lambda_0}{\lambda_{\text{Ra}}}$ is close to 1. In this case, the Ångström exponent å does not affect the extinction coefficient solution. Conversely, when the extinction coefficient is derived from the vibrational-rotational Raman signals with $\lambda_{\text{Ra}} = 607$ nm, the ratio $\frac{\lambda_0}{\lambda_{\text{Ra}}}$ is smaller than 1 (specifically, 0.876). As a result, the uncertainty in the Ångström exponent input value causes an uncertainty in the calculated extinction coefficient.

## 2.3 Particle backscatter coefficient

The equation of the backscatter coefficient $\beta$ (Ansmann et al., 1992), using a Raman lidar, is given as follows:

$$\beta_{\lambda_0}^{\text{par}}(R) = \left[\beta_{\lambda_0}^{\text{par}}(R_0) + \beta_{\lambda_0}^{\text{mol}}(R_0)\right] \frac{P_{\lambda_0}(R)P_{\lambda_{\text{Ra}}}(R_0)N(R)}{P_{\lambda_{\text{Ra}}}(R)P_{\lambda_0}(R_0)N(R_0)} \times \frac{\exp\left[-\int_{R_0}^{R}\left(\alpha_{\lambda_{\text{Ra}}}^{\text{par}}(r) + \alpha_{\lambda_{\text{Ra}}}^{\text{mol}}(r)\right) dr\right]}{\exp\left[-\int_{R_0}^{R}\left(\alpha_{\lambda_0}^{\text{par}}(r) + \alpha_{\lambda_0}^{\text{mol}}(r)\right) dr\right]} - \beta_{\lambda_0}^{\text{mol}}(R). \tag{2}$$

The Ångström exponent influences the backscatter solution via the terms $\alpha_{\lambda_0}^{\text{par}}(r)$ and $\alpha_{\lambda_{\text{Ra}}}^{\text{par}}(r)$ which are connected by the assumed Ångström exponent, i.e., $\alpha_{\lambda_{\text{Ra}}}^{\text{par}}(r) = \alpha_{\lambda_0}^{\text{par}}(r)\left(\frac{\lambda_0}{\lambda_{\text{Ra}}}\right)^{\text{å}}$ is included in the backscatter coefficient computation. Additionally, the reference height $R_0$ and the direction of integration (forward or backward) also influence the retrieval. Therefore, the Ångström exponent influence on the backscatter coefficient is more complex. The impact of å on $\alpha_{\lambda_0}^{\text{par}}(R)$ and $\beta_{\lambda_0}^{\text{par}}(R)$ will be shown in 3. In the case studies presented here, the starting height is set at the far end range, $R_0 > R$.

## 2.4 Lidar ratio

The lidar ratio S is defined as the ratio of the extinction-to-backscatter coefficient (Ansmann et al., 1992).

$$S = \frac{\alpha}{\beta} \tag{3}$$

In Sect. 2.1 and 2.2 we demonstrated the impact of the Ångström exponent on the backscatter and extinction coefficients retrieved by using the vibrational-rotational Raman lidar solution. Both coefficients are influenced by the Ångström exponent, which in turn affects the lidar ratio in an even more complex way.

## 3 Results

To investigate the impact of the assumed Ångström exponent on the solutions for the extinction coefficient, backscatter coefficient, and lidar ratio when using rotational-vibrational Raman signals, the respective rotational Raman lidar solutions are used as reference. Three different aerosol scenarios and a cirrus case are discussed in detail. The first case examines a wildfire smoke layer, while the second focuses on a Saharan dust event. The third case considers residential wood combustion smoke, and the final deals with a cirrus observation. For all these cases, the profiles of the backscatter coefficient $\beta$, depolarization ratio $\delta$, extinction coefficient $\alpha$, and lidar ratio S are shown with the respective error bars. The error bars, mainly the result of signal noise, indicate uncertainties of 10% in the case of $\beta$ (BSC) and $\delta$ (PDR), 20% and 25% in the case of $\alpha$ (EXT) and S (LR), respectively. Figure. 2 shows HYSPLIT (Hybrid Single Particle Lagrangian Integrated Trajectory ) air-mass backward trajectories for the first three case studies. (HYSPLIT, 2025; Stein et al., 2015; Rolph et al., 2017)

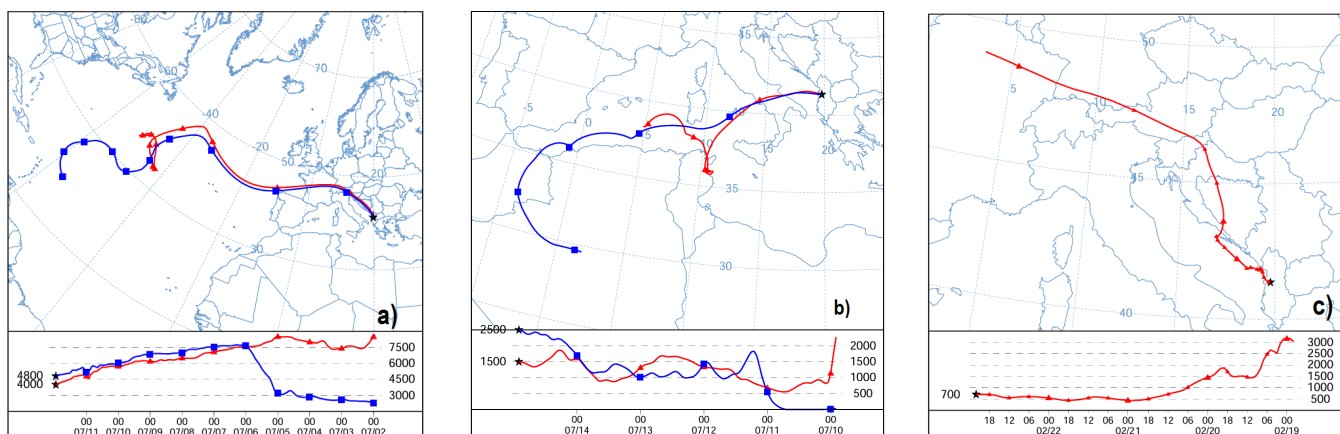

**Figure 2.** HYSPLIT backward trajectories arriving at Tirana, Albania (a) on July 11, 2023 (10 days), (b) July 14, 2023 (5 days), (c) and February 22, 2023 (4 days). The lidar observations of (a) wildfire smoke, (b) Saharan dust, and (c) residential wood burning smoke are discussed in Sects. 3.1, 3.2, and 3.3, respectively.

### 3.1 Case study: Wildfire smoke

A smoke layer was observed with the Raman lidar in Tirana, Albania, at altitudes between 3 and 5 km height on July 11–12, 2023. The 10-day backward air mass trajectories in Fig. 2a indicate an intercontinental long-range transport of wildfire smoke, most probably from the USA and Canada.

The backscatter coefficient peaked at 5 $\mathrm{Mm}^{-1}\mathrm{sr}^{-1}$. The depolarization ratio remained low with values below 5%. The extinction coefficient within this smoke layer reached a maximum of 300 $\mathrm{Mm}^{-1}$, and the corresponding lidar ratio showed values from 60-70 sr which are typical for aged wildfire smoke (Haarig et al., 2018).

The optical properties, except the depolarization ratio, are retrieved by using both rotational and vibrational-rotational Raman signals for different Ångström exponents. Generally, an increase in the Ångström exponent value shifts the extinction

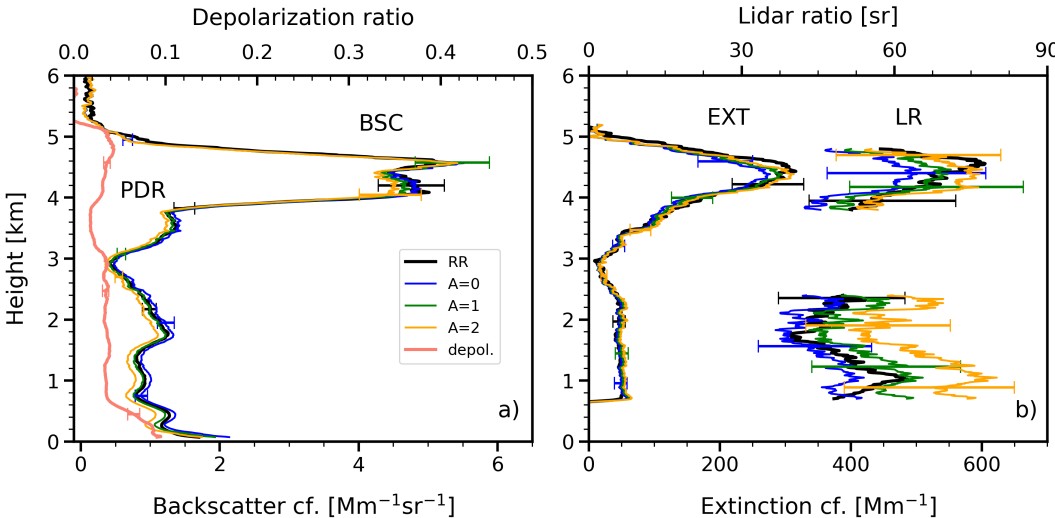

**Figure 3.** Profiles of aerosol optical properties: a) backscatter coefficient (BSC) and particle depolarization ratio (PDR); b) extinction coefficient (EXT) and lidar ratio (LR) measured from 11 July 2023 at 20:40 to 12 July 2023 at 01:30 UTC, when an aged wildfire smoke layer between 3 and 5 km height crossed the lidar station. In the computation of the optical properties from the vibrational-rotational Raman signals, we assumed an Ångström exponent A (see legend in panel a) of 0 (blue), 1 (green), and 2 (orange). The respective rotational Raman lidar solutions (RR) are given as thick black lines. Error bars show the retrieval uncertainty mainly caused by signal noise.

coefficient and the lidar ratio to slightly larger values and the backscatter coefficient to lower values. The increase of the assumed A value from 1 to 2 results in a decrease in the median backscatter coefficient in the smoke layer at heights >4 km from about $4.5\,\mathrm{Mm^{-1}sr^{-1}}$ to $4.3\,\mathrm{Mm^{-1}sr^{-1}}$. In contrast, the extinction coefficient increases from about $215\,\mathrm{Mm^{-1}}$ to $230\,\mathrm{Mm^{-1}}$, whereas the lidar ratio rises from around 65 to 70 sr. Thus, the A-assumption-related relative uncertainty is of the order of <5% (backscatter), about 7% (extinction), and almost 10% (lidar ratio) in this case when we assume an A value of 1 and the true Ångström exponent is 0 or 2. The comparison of the rotational and vibrational-rotational Raman lidar solutions for the extinction coefficient and the lidar ratio suggests that the true Ångström exponent was between 1 and 2.

## 3.2 Case study: Saharan dust

The air mass trajectories in Fig. 2b indicate long-range transport of Saharan dust at 2.5 km height to Albania within 5 days (10-14 July 2023). The lidar observations in Fig. 4 show two layers containing dust. In the first layer from 1 to 1.7 km height, the depolarization ratio of 20-25% indicates a mixture of dust and anthropogenic particles. The lidar ratio is in the range of 35-50 sr. The comparison of the rotational and vibrational-rotational Raman lidar solutions for the extinction coefficient and the lidar ratio suggests a true Ångström exponent of 1-2.

Between 2 and 3 km height, a strong Saharan dust layer is then observed, showing a maximum backscatter coefficient of about $3.5\,\mathrm{Mm^{-1}sr^{-1}}$ and a particle depolarization ratio close to 30% in the layer center. In the layer center, the different Raman

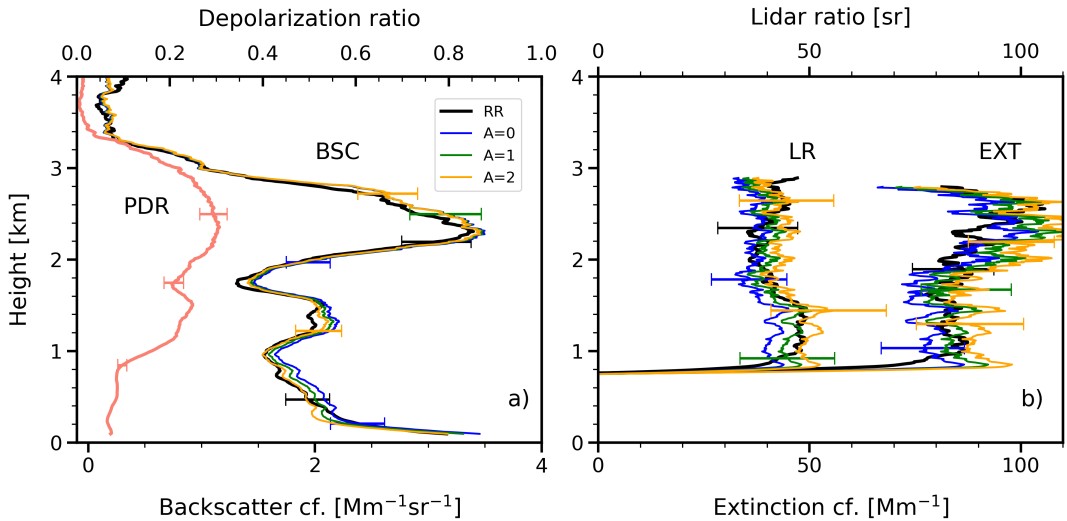

**Figure 4.** Same as in Fig. 3 except for the 14 of July, 20:10-22:59 UTC. A strong Saharan dust layer crossed the lidar station between 2 and 3 km height.

lidar solutions for the lidar ratio and the extinction coefficient agree best for A=0, as expected for coarse-mode-dominated dust particle size distributions.

When we increase the A input value from 0 to 1, the layer median value of the backscatter coefficient decreases from about 2.93 $\mathrm{Mm}^{-1}\mathrm{sr}^{-1}$ to 2.90 $\mathrm{Mm}^{-1}\mathrm{sr}^{-1}$ in the case of the vibrational-rotational Raman lidar solutions, while the extinction coefficient and lidar ratio increase from about 90 $\mathrm{Mm}^{-1}$ to almost 100 $\mathrm{Mm}^{-1}$ and from 37 sr to 40 sr, respectively. Again, the impact of the A input on the vibrational-rotational Raman lidar solution is low with relative uncertainties of about 1% (backscatter) and 7-10% (extinction, lidar ratio).

### 3.3   Case study: Residential wood combustion smoke

According to HYSPLIT's 4-day backward trajectories in Fig. 2c, slowly transported air masses accumulating pollution from the Balkans region reached Tirana at an altitude of 700 m on February 22, 2023. A thick aerosol layer with a maximum backscatter coefficient of 3.5 $\mathrm{Mm}^{-1}\mathrm{sr}^{-1}$ was observed at 700 m on this late winter day. The dominating aerosol type was smoke from residential wood combustion. The smoke-polluted aerosol layer extended up to 1.4 km height. The lidar ratio was about 90 sr and maximum extinction coefficients close to 350 $\mathrm{Mm}^{-1}$. The very low depolarization ratio indicated spherical particles.

    With increasing A from 0 to 2 the backscatter coefficient decreases, bringing the backscatter coefficient profile closer to the respective backscatter profile retrieved from the rotational Raman signals (thick black line). In contrast, the extinction coefficient and lidar ratio show an opposite trend. With increasing A from 1 to 2 within the main layer (600-1100 m height), the extinction coefficient and lidar ratio increase from about 275 $\mathrm{Mm}^{-1}$ to 295 $\mathrm{Mm}^{-1}$ and from around 85 sr to 95 sr, respectively. Again, for A of 1-2, the vibrational-rotational Raman lidar solutions agree well with the respective rotational Raman lidar

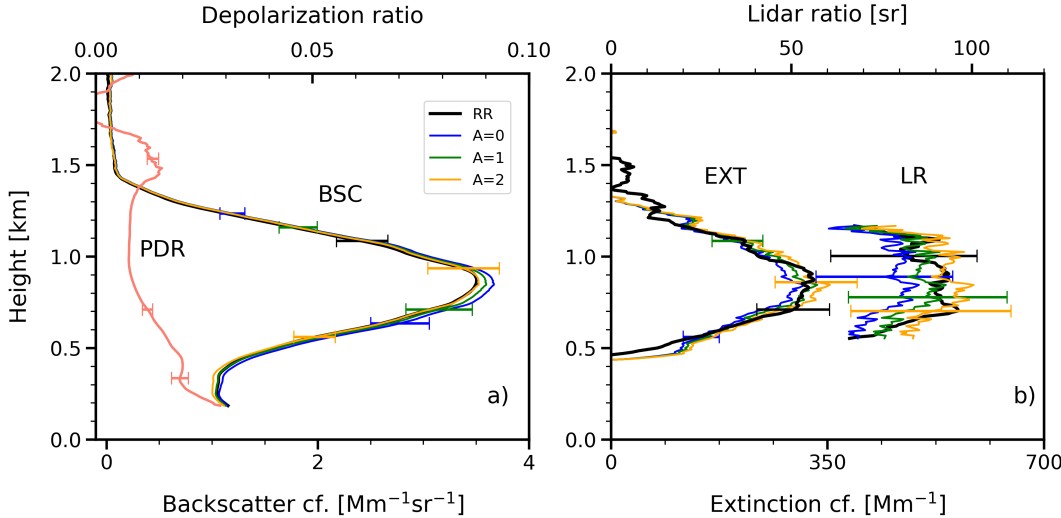

**Figure 5.** Same as in 3, except for the 22 February 2023, 21:30-22:30 UTC. On this late winter day smoke originating from residential wood combustion heavily polluted the air in Tirana.

solutions, as expected for typical smoke size distribution with a pronounced accumulation mode. In terms of A-assumption-related relative uncertainty, we obtain again errors of the order of a few percent (backscatter), between 5 and 10% (extinction) and around 10% (lidar ratio) when the true and assumed A values deviate by 1.

### 3.4 Case study: Cirrus

A cirrus observation is presented in Fig. 6. Ice crystal observations can be regarded as ideal validation experiments. Because ice crystals are much larger compared to the laser wavelength, ice crystal scattering is wavelength independent (A=0) and both solutions for the extinction coefficient determined from the rotational and vibrational-rotational Raman lidar observations (with A=0) should match perfectly. As can be seen in Fig. 6, this is the case for the lower part of the cirrus (8-9.5 km height) when we keep the impact of signal noise, indicated by the error bars, into account. Above 10 km height, the extinction profiles

were no longer useful for comparison. Strong attenuation by ice crystal scattering drastically reduced the signal-to-noise ratio and correspondingly increased the statistical uncertainty. In addition, multiple scattering affected the observations, especially in the upper part of the cirrus layer (above 11 km height).

### 4 Conclusions

We investigated the impact of the assumed Ångström exponent on the full set of aerosol products (backscatter and extinc-
tion coefficients and lidar ratio) that can be determined from nitrogen vibrational-rotational Raman lidar observations. The solutions were compared with respective products obtained by using the pure rotational Raman lidar technique, in which an

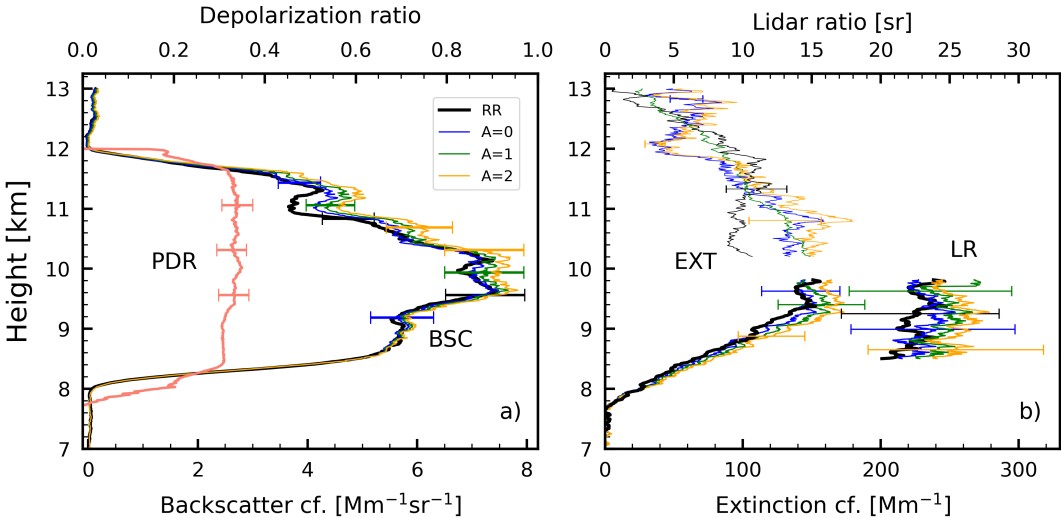

**Figure 6.** Same as in Fig. 3 except for the 2 January, 19:00–23:59 UTC. A vertically thick (8-12 km) and horizontally extended cirrus field was monitored with the Raman lidar. The vertically constant depolarization ratio of around 35% indicates a homogeneous ice crystal size and shape characteristics.

Ångström exponent is not needed as an input parameter. We found that the impact of a wrong Ångström exponent on the determined aerosol optical properties is low, i.e., causing relative uncertainties of the order of 3-10%. From the point of view of a careful characterization of aerosol optical properties, both Raman lidar approaches can thus be well applied. Retrieval

uncertainties are low. By combining both Raman lidar methods, there is even the chance to characterize the wavelength dependence of aerosol backscattering and extinction in the visible wavelength spectrum (500-650 nm) from single-wavelength 532 nm lidar observations, as our studies show. An Ångström exponent of 0–0.5 is recommended for pure or polluted dust. Ångström exponent values of 1.0 and 1.5 are more appropriate for non-dust aerosols, such as anthropogenic pollution. In the case of wildfire smoke, it is more complex. For very aged wildfire smoke, the extinction-related A approaches 0.0, while the

backscatter-related A is 1.5, resulting in higher lidar ratios at 532 nm than at 355 nm, as observed for Canadian (Haarig et al., 2018), Australian (Ohneiser et al., 2020), and Siberian (Ohneiser et al., 2021) smoke. For black carbon-rich aerosols, such as those from residential wood combustion, the A value can be set between 1.5 and 2.0. As a final remark and for completeness, one should mention that the rotational Raman lidar signals allow even measurements up to several kilometers in height in daytime conditions because the intensity of rotational Raman backscattering by oxygen and nitrogen molecules is more than an

order of magnitude larger than the intensity of nitrogen vibrational-rotational backscattering (Zenteno-Hernández et al., 2021).

*Data availability.* Polly_1v2 lidar data are stored in the PollyNET database with quicklooks at http://polly.tropos.de/. All analyzed products are available upon request at www.polly@tropos.de. Backward trajectory analysis was conducted using air mass transport computations

with the NOAA (National Oceanic and Atmospheric Administration) HYSPLIT (HYbrid Single-Particle Lagrangian Integrated Trajectory) model HYSPLIT (HYbrid Single-Particle Lagrangian Integrated Trajectory) model (http://ready.arl.noaa.gov/HYSPLIT_traj.php, HYSPLIT, 2025).

*Author contributions.* GM analyzed the data together with HB, CJ, JH, and NS and wrote the manuscript. AA conceptualized the study, supported the discussion of the results, and edited and reviewed the manuscript. RE and SSh supported the measurements with Polly_1v2.

*Competing interests.* The authors declare that they don't have competing interests.

*Acknowledgements.* The authors thank the Leibniz Institute for Tropospheric Research (TROPOS) group for their effort in conducting lidar measurements in Tirana, Albania, and for their great support with supervising and data analysis.

*Financial support.* This work is performed in the frame of doctoral studies. I express my gratitude to DBU (Deutsche Bundesstiftung Umwelt) (Grant No. 30024/032) and GIZ (Deutsche Gesellschaft für Internationale Zusammenarbeit) (Agreement No. 81281212, Project No. 17.2192-7.003.00) for supporting my Ph.D. research through scholarships.

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
