# Peer review of "Ångström exponent impact on the aerosol optical properties obtained from vibrational-rotational Raman lidar observations"

_EGUsphere, 2025_

## Author Response (AR2)

Manuscript considers the influence of the Angstrom exponent on calculation of the backscattering and extinction coefficients from measurements of Mie-Raman lidar. Several measurement cases related to different types of aerosol are considered.

This subject is not new, and is well familiar to every researcher working with Mie-Raman lidar. On another hand, it is useful to summarize it and to provide the expected errors for different aerosol types. Thus, from my point of view, the manuscript is suitable for AMT.

Authors work in this field for long time and are good experts, so I have not much to add. Just several technical comments.

Authors: We thank the reviewer for the positive feedback and are pleased that the manuscript is considered suitable for AMT. We have carefully addressed the technical comments in the revised version.

Reviewer: p.5 ln.117 "In general, as the Ångström exponent A increases, the extinction coefficient…"

Should be reformulated.

Authors: This sentence is reformulated (page 6, line 117-118): Generally, an increase in the Ångström exponent value shifts the extinction coefficient and the lidar ratio to slightly larger values and the backscatter coefficient to lower values.

Reviewer: Fig.3. Extinction at 3 km height is low and drop of the lidar ratio at 3 km probably has no physical meaning, So, by my opinion, no reason to show it. Within aerosol layer at 4 km, backscattering and extinction coefficients have very different profiles. And looks like extinction is strongly smoothed. Thus, profile of lidar ratio, probably, makes sense only in the center of the layer. I would show only averaged value (near the center of the layer) of the lidar ratio.

Authors: We thank the reviewer for the recommendation. Since we aim to highlight the profile variations and deviations among different Ångström Exponents, we believe it is more informative to present the profiles rather than only mean values. However, we agree that the drop in the LR at 3 km has no physical meaning. We decided to cut the LR profiles at around 3 km and exclude them from the plot.

Figure 3 has been revised accordingly.

[Figure]

Reviewer #2

The authors perfrom a sensitivity study concerning the impact of the assumed Angstrom Exponent, usually chosen as 1, on the Raman lidar invresions. Their lidar system, combining vibrational and rotation Raman channels allows to use as a reference retrieval the one from the rotational channel, which makes feasible such a sensitivity study. The approach is simple and clear, well presented and documented. The results are useful for the lidar community especially for the estimation of the uncertainties in the retrievals from the vabrational Raman channels. The only comment I have is, that I would expect in the conclusions, apart from the quantification of the uncertainties, a recommendation for the choice of the Angstrom exponent in the operatinal processing of lidar measurements.

Authors: We thank the reviewer for the supportive evaluation and appreciate the positive comments on the clarity, usefulness, and relevance of our work.

We added the following paragraph (page 9, line 172-177): An Ångström Exponent (AE) of 0–0.5 is recommended for pure or polluted dust. AE values of 1.0 and 1.5 are more appropriate for non-dust aerosols, such as anthropogenic pollution. In the case of wildfire smoke, it is more complex. For very aged wildfire smoke, the extinction-related AE approaches 0.0, while the backscatter-related AE is 1.5, resulting in higher lidar ratios at 532 nm than at 355 nm, as observed for Canadian (Haarig et al., ACP, 2018), Australian (Ohneiser et al., ACP, 2020), and Siberian (Ohneiser et al., ACP, 2021) smoke. For black carbon-rich aerosols, such as those from residential wood combustion, the AE value can be set between 1.5 and 2.0.